# Analysis of blood parameters and molecular endometrial markers during early reperfusion in two ovine models of uterus transplantation

Marie Carbonnel[1,2,3]☯*, Nathalie Cornet[1,2,3]☯, Aurélie Revaux[1,2], Angéline Favre-Inhofer[1,2,3], Laurent Galio[2,3], Mariam Raliou[2,3], Anne Couturier-Tarrade[2,3], Corinne Giraud-Delville[2,3], Gilles Charpigny[2,3], Valérie Gelin[2,3], Olivier Dubois[2,3], Barbara Hersant[4], Romain Bosc[4], Raphael Coscas[5,6], François Vialard[2,3], Pascale Chavatte-Palmer[2,3], Christophe Richard[2,3], Olivier Sandra[2,3]☯, Jean-Marc Ayoubi[1,2]☯

1 Department of Gynaecology and Obstetrics, Foch Hospital, Suresnes, France, 2 Université Paris-Saclay, UVSQ, INRAE, BREED, Jouy-en-Josas, France, 3 Ecole Nationale Vétérinaire d'Alfort, BREED, Maisons-Alfort, France, 4 Department of Plastic, Reconstructive, Aesthetic and Maxillofacial Surgery, Henri Mondor Hospital, Créteil, France, 5 Department of Vascular Surgery, Ambroise Paré University Hospital, Boulogne-Billancourt, France, 6 UMR 1018, Inserm-Paris11 - CESP, Versailles Saint-Quentin-en-Yvelines University, Paris-Saclay University, Boulogne-Billancourt, France

☯ These authors contributed equally to this work.
* m.carbonnel@hopital-foch.com

**Data Availability Statement:** All relevant data are within the manuscript and its Supporting information files.

## Abstract

The dissection of the veins is the trickiest step of Uterine transplantation (UTx). Performing the anastomosis of a single uterine vein could bring a therapeutic benefit and simplification of surgery and serve for managing unilateral venous thromboses. The objectives of this project were to evaluate the expression of early markers of ischemia-reperfusion and to compare findings following one or two vein anastomoses. Orthotopic uterine auto-transplantations were performed on an ovine model with anastomosis of either two (group 1) or one utero-ovarian veins (group 2). Blood gases, histology and ischemia- reperfusion markers transcripts (*PTGS2*, *IL6*, *IL8*, *SOD2*, *C3*, *BAX/BCL2* and *TLR4)* were analyzed as well as PTGS2 protein expression using Western Blot and fluorescence immunolocalization on endometrial biopsies after 3h of reperfusion. Ten ewes were included in the experimentation, 4 were in group1, 3 in group 2, the others being sham operated controls. No significant differences were observed between the two phenotypes. Based on these results, the anastomosis of one single uterine vein appears to be an approach consistent with short-term graft survival. Further experiments will be needed to confirm the reliability of this approach, especially the long-term follow-up of the uterine graft including its ability to support gestation to term.

## Introduction

In vitro fertilization (IVF), developed over 40 years ago for treating tubal infertility, today is used to treat nearly all infertility issues. However, one cause of infertility remains untreated: uterine factor infertility (UFI). UFI, which is estimated to affect 1–5% of women, can be

**Funding:** The authors received funding (financial and material support) from the Foch Foundation. Angéline Favre-Inhofer and Nathalie Cornet were paid a stipend by the Foch Foundation. The funders had no role in study design, data collection and analysis, decision to publish, or preparation of the manuscript.

**Competing interests:** The authors have declared that no competing interests exist.

congenital (complete uterine agenesis or other severe congenital malformations) or acquired (from hysterectomy or loss of a functional endometrium) [1]. Since the initial report of the first case of successful pregnancy in a transplanted uterus by Brannström's team in 2014 [1], more than 70 cases of uterine transplantation (UTx) have been performed across the world with 53 reported in peer reviewed journals, mostly with live donors (LD) [2–15]. More than 20 healthy children were born after UTx [1,16–20]. For LD, the uterus retrieval requires extensive surgery in order to preserve enough uterine vessels length (usually uterine arteries and veins) to enable end-to-side anastomoses with the external iliac vessels of the recipient. Dissection of uterine veins is very complex, leading to long operative times—10 hours or more—in case of LD [12] and exposing to significant surgical risks notably, involving lesions to the ureters [5,21–23]. For the recipient, the anastomosis of vessels represents another challenging step because of small vessel size that lead to long warm ischemia times and exposes to the risk of explantations in case of thrombosis (20 to 30% of the cases) [7].

The uterus is the only organ with two major arteries (left and right uterine arteries) and four major venous outflows (2 uterine veins and 2 utero-ovarian veins). So far, 2 arteries and 2 veins have been used in UTx [7]. In other solid transplanted organs, natural vascularization involves mainly one large artery and one large vein. Therefore, there is no comparable model of solid organ transplantation in the literature for which only one venous outflow rather than two or more could suffice for the quality of the grafted organ. The uterus has a large anastomotic network that may enable the preservation of vessels necessary for appropriate perfusion [24]. Indeed, if two arteries are essential, venous outflow can be supported by one unique vein. Furthermore, ischemia-reperfusion in solid organ transplantation (SOT), which depends on the number and the size of vessels anastomosed, is responsible for organ dysfunction and increases the incidence of acute and chronic rejection [25–27]. The birth of one cynomolgus macaque has been reported with only one functional venous outflow [28] No additional data are available. Therefore, using only one vein for outflow or avoiding graft explantation when a thrombosis is observed in one of the two veins could be a valuable option in order to reduce complications for LD and warm ischemia times [29,30].

Sheep is the most suitable large animal model of UTx due to comparable sizes of the uterus and blood vessels as well as singleton pregnancy [31]. Two anatomic differences have to be mentioned: in ovine, the uterus is bicornuate and only utero-ovarian veins quite similar to ovarian vein in humans are available for venous outflow. Auto-transplantation was chosen in this study to avoid bias induced by early rejection.

The objective of this project was to provide a first insight into the influence of one or two vein anastomosis on the early steps of ischemia-reperfusion in an ovine model of uterine transplantation. We analyzed blood gas (pH, pO2/pCO2 and lactates), over a 3 hour period post-reperfusion, as usually performed to evaluate early ischemia-reperfusion injuries in animal UTx [31]. Using histological sections of uterus, we carried out stereological analyses and assessed neutrophil numbers. In the endometrium, we quantified transcripts levels of a selection of genes known to be involved in inflammation, apoptosis and immunity (*BAX*, *BCL2*, *C3*, *IL6*, *IL8/CXCL8*, *PTGS2*, *SOD2*, *TLR4*) and analyzed the expression as well as the cell localization of a major inflammation-related protein (PTGS2) by Western-blot and immunofluorescence respectively.

## Materials and methods

### Ethical approval

Experimental procedures were approved by the INRAE-AgroParisTech animal ethics committee (CEEA 45) with the authorization subsequently granted by the French Ministry of National

Education,Teaching and Research (apafis#13118–2017122219387116_v1). All animals were treated according to the directive 2010/63/eu of the European parliament relative to the protection of animals and compliant with the ARRIVE criteria.

## Animals

Cycling biparous ewes (Romane breed) were synchronised using a vaginal progesterone pessary placed for 10 days before surgery to normalize progesterone levels at the time of surgery. Median age was 38 months (Inter-quartile (IQ) 35.5–41.5) and median weight was 76.8 kg (IQ 72.5–83.5). Three ewes had a simple hysterectomy immediately after anaesthesia and constituted the control group. For the uterus autotransplantations, the ewes were brought to the operating theatre on the day of the experiment one by one. A randomization between group 1 (anastomosis of 2 veins) and group 2 (anastomosis of 1 vein) was performed on backtable after the retrieval of the uterus and before the auto-transplantation. All surgeries were performed between May 2018 and February 2019.

## Anaesthesia

Sedation and general anaesthesia were induced with butorphanol (0,1 mg/kg) (Torbugesic®, Zoetis, France) intramuscular; diazepam (0,5 mg/kg) (Diazepam, TVM, France) and ketamine (4 mg/kg) (Imalgene 1000, Boehringer Ingelheim Animal Health, France) intravenously (i.v.). After orotracheal intubation, anesthesia was maintained by inhalation of a gas mixture of 1–2% isoflurane with air/oxygen. Ruminal contents were drained through a stomach tube. The urinary bladder was catheterized via the urethra. Throughout the surgery, an intravenous line was inserted in jugular to compensate fluid loss with Ringer lactate solution (Viaflo, Baxter, France). A volume expander (Hydroxyethyl starch 130/0.4, Restorvol®, Octapharma, France) was administered when bleeding was more than 500 ml or when the surgery lasted more than ten hours. At the end of the experiment each ewe was euthanized with an i.v. injection of pentobarbital (20 ml). (Dolethal®, Vétoquinol, France).

## Surgery

The surgical technique was described in our previous study [32]. The different steps of the surgical protocol are summarized in Fig 1. Briefly, the skin was opened from the pubic bone extending to a length of about 25 cm. For exposure of the uterus, the intestinal loops were mobilized and retracted. The uterine artery and vein were identified on both sides and further dissected to their respective origins from the internal iliac vessels. Care was taken to avoid injury of the ureter (Fig 2a). After isolation of the uterus, the vagina was cut by monopolar diathermy. Vascular clamps were placed on the origin of the uterine arteries and utero-ovarian veins after i.v. injection of 5 000 international units (IU) of heparin, and all vessels were cut in the same sequence. The uterus was placed on ice and flushed with heparinized lactated Ringer's solution (5000 IU in 500 ml of Ringer's solution) to obtain a clear outflow while the origin of the vessels was ligated (Fig 2b). In group 2 the most suitable vein was chosen. End-to-side anastomoses between uterine arteries and external iliac arteries and utero-ovarian veins and external iliac veins were performed after injecting a second dose of 15,000 IU of heparin i.v. (Fig 2c).

Animals were included in the study only when all anastomoses were functional with (i) recoloration of uterus, (ii) positive patency test, and (iii) vascular flow observed with ultrasounds in uterine arteries during 3 hours after reperfusion and post-operative dissection without thrombosis or transfixion point (Fig 2d). Blood gas analyses (I-Stat 1, Abbott) were analyzed in one utero-ovarian vein before, immediately after reperfusion (M0) (Minute), or 15

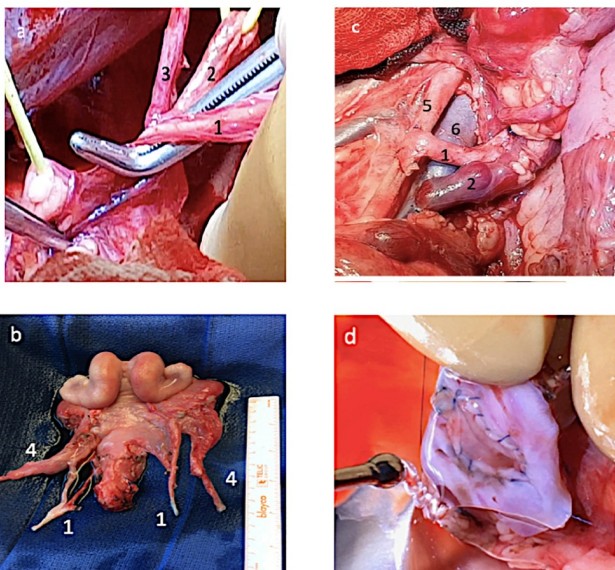

**Fig 1. The different steps of the surgical protocol.** D-10: Synchronization of the estrous cycle by an intra-vaginal progesterone sponge D0: Day of surgery; T: Time in minutes.

minutes (M15), 30 minutes (M30), 60 minutes (M60), 120 minutes (M120) and 180 minutes (M180) after reperfusion. Lactates, $pCO_2$, $pO_2$, and pH were also quantified. Eventually, the uterus was collected before the ewe was euthanized. Dissection of anastomoses was performed to confirm the patency. In the control group, a simple hysterectomy without dissection was carried out, then the uterus was collected without delay.

Total operative time accounts for the time between the first incision and the hysterectomy, including the 3 hours reperfusion. Warm ischemia was divided in two successive parts: first, the time between the clamping of the uterine vessels and the retrieval of the uterus; and second, the time between the beginning of the anastomosis and uterine reperfusion. Cold ischemia refers to the time of backtable flushing on ice.

## Uterine dissection and sample collection

The two horns were treated separately. Each horn was opened longitudinally. Caruncular (CAR) and inter-caruncular (ICAR) endometrial samples were dissected separately from the upper, middle and lower part of each horn. Middle part were used for gene expression. Fresh

**Fig 2. Photographs showing various steps of uterus auto-transplantation.** a) Dissection between uterine vessels and ureter; (b) Excised uterus on the backtable; (c) anastomosis; (d) uterine anastomosis at dissection. 1 = uterine artery; 2 = ureter; 3 = umbilical artery; 4 = utero-ovarian vein; 5 = external iliac artery; 6 = external iliac vein.

**Fig 3. Histology of the upper part of a uterine horn in sheep.** (a) Hematoxylin–eosin stained section of the upper horn. Arrows delineate the various tissues of the uterine wall (X20 magnification used for stereology). (b) Hematoxylin-eosin stained section of the upper horn (X40). Arrows indicate infiltrating neutrophils. S, Stroma; G, endometrial Gland; A, Artery; M, Muscle; MCT, Muscular Connective Tissues; LE, Luminal Epithelium and OE, Oedema.

tissue was immediately frozen in liquid nitrogen then stored at -80˚C before subsequent real-time PCR analyses from total RNA and Western Blot protein assays.

Samples of the lower and upper portions of the horns encompassing the entire uterine wall (serosa, myometrium and endometrium) were fixed in a 1X PBS solution containing 4% para-formaldehyde before being included in paraffin and in Tissue-Tek® OCT cryo-embedding compound (Sakura Finetek, France) and were stored at 4˚C and -80˚C, respectively, until use for histological analyses and immunolocalization.

## Histology

Lower and upper portions of the horns were sectioned (3μm slides) and stained with hematoxylin-eosin. The sections were analyzed by light microscopy for sign of ischemic injury. Oedema, stasis, necrosis and apoptosis were noted for each horn in lower and upper portions. Neutrophils density was quantified by counting tissue bound neutrophils in 10 area that were randomly chosen (40 fold magnification), with the use of a grid (Fig 3). Sections were also scanned using Nano-Zoomer Digital Pathology® (Hamamatsu Photonics) to perform stereology. Stereology is a quantitative method where measurements made on 2 dimensions sections are compiled to infer quantitative data on the 3-dimension structure of the studied organ [33]. For each uterus, upper portions of the horn were analyzed at a 20-fold magnification. For each upper portion of the horn, around 30 linear dipole probes were randomly distributed over the section. Components crossed by the linear dipole were computed as luminal epithelium, stroma, endometrial gland, muscle, muscular connective tissues, perimetrium, veins, arteries and oedema (Fig 3). Their relative volume (Vv,%) were calculated using the histological Quantification and mapping *One-Stop Stereology* (MercatorPro®, ExploraNova) [33].

## Quantification of gene expression in the endometrium

Total RNAs were extracted from endometrial CAR and ICAR tissue samples from the middle part of the uterine horns using Trizol® Reagent (Ambion®, Life Technologies, Les Ulis, France) and Ultra-Turrax dispensers (IKA-Labortechnik) as previously described [34]. Total RNA samples were purified on Qiagen columns following the manufacturer's protocol including a DNase treatment step (RNeasy Mini kit; Qiagen, Courtaboeuf, France). Reverse transcription was performed using 1 μg total RNA, Oligo (dT)18 primer (final concentration: 30 ng/μL, Thermo Fisher Scientific, Illkirch, France), dNTPs (final concentration: 10 mM each, Thermo Scientific) and Maxima H Minus (reverse transcriptase, final concentration: 10 U/μL, Thermo Fisher Scientific, Illkirch, France). Primers used are described in Table 1. The reverse transcription conditions were as follows: 65˚C for 5 min, 50˚C for 30 min and 85˚C for 5 min. Each qPCR reaction was carried out using Step One Plus™ Real-Time PCR System and SYBR™

**Table 1. Primer sequences.**

| Gene | Encoded protein | Accession N° | Primer sequence (forward; reverse) | Amplicon (bp) |
|------|----------------|--------------|-----------------------------------|---------------|
| *BAX* | BCL2 associated X protein | NM_173894 | CTCCCCGAGAGGTCTTTTTC; TCGAAGGAAGTCCAATGTCC | 176 |
| *BCL2* | B cell lymphoma 2 protein | NM_001314213 | TCGTGGCCTTCTTTGAGTTC; CGGTTCAGGTACTCGGTCAT | 109 |
| *C3* | C3 complement | XM_027969774 | AGAAGCAGAAGCCTGATGGA; CCTCGCAGATGTCTTTAGCC | 150 |
| *C2ORF29* | CCR4-NOT transcription complex subunit 11 | XM_002691150 | CCTTCAAGAGCCCCCTGT; GGGTCCTTTTCCAACTCTCC | 64 |
| *GAPDH* | Glyceraldehyde-3-phosphate dehydrogenase | NM_001034034 | GCTGACGCTCCCATGTTTGT; TCATAAGTCCCTCCACGATGC | 151 |
| *IL6* | Interleukine-6 | NM_001009392 | TAACCACTCCAGCCACACAC; GCGTTCTTTACCCACTCGTT | 70 |
| *IL8/CXCL8* | Interleukine-8 | NM_001009401 | TGTGTGAAGCTGCAGTTCTGT; TGGGGTCTAAGCACACCTCT | 186 |
| *PTGS2* | ProastaglandinSynthase-2 | NM_001009432 | TCCGCCAACTTATAATGTGCAC; GGCAGTCATCAGGCACAGGA | 101 |
| *SOD2* | Superoxyde Dismutase-2 | NM_001280703 | GGATCCCCTGCAAGGAACAA; TGGCCTTCAGATAATCGGGC | 110 |
| *TLR4* | Toll Like Receptor 4 | | GACCCTTGCGTACAGGTTGT; GGCTGCCTAAATGTCTCAGGT | 136 |

Bp: Base pairs.

Green PCR Master Mix(Applied Biosystems®, Villebon sur Yvette, France). Assays were performed in duplicate for each sample. The reaction mixture for amplification contained 5 μL cDNA diluted 1:200 in a final reaction volume of 15 μL. The PCR program involved two initial stages at 50 ˚C for 2 min and 95 ˚C for 10 min, followed by 45 cycles at 95 ˚C for 15 s and a final annealing and extension step at 60 ˚C for 1 min. Amplified PCR fragments were sequenced to assess the amplification of the correct fragment. To confirm product specificity, melting-curve analyses were also performed immediately after amplification (from 60 to 95˚C in steps of 0.8˚C). Standard curve was included for each gene to generate arbitrary expression values for candidate genes. The Qbase software (Biogazelle,) based on the geNorm algorithm was used for analyzing quantitative PCR data. The geNorm module allows the determination of the most stable reference (housekeeping) genes from a set of tested candidate reference genes in a given sample panel. Glyceraldehyde-3-phosphate dehydrogenase (*GAPDH*) and CCR4-NOT transcription complex subunit 11 (*C2ORF29*) were determined as housekeeping genes. All gene expression data for the transcripts of interest were expressed as mean calibrated normalized relative quantity (CNRQ) values in arbitrary units.

**Western-blot analysis of PTGS2 protein expression.** Endometrial ICAR tissue from middle part of uterine horns was ground with an Ultra-Turrax T25 grinder (IKA-Labortechnik) in a buffer solution allowing cell lysis and protein denaturation. Extraction buffer was prepared with Tris (0.02M)/EDTA (0.05M)/NaCl (0.15M) (pH 7.6), protease inhibitors (Protease inhibitor Cocktail, Sigma Aldrich, 20 μl/sample). In a second step, the samples were prepared for gel migration. After centrifugation for 30 minutes at 14,000 x*g* the protein precipitate was collected. Proteins were reduced and homogenized in a Laemmli 1X buffer with β-Mercaptoethanol solution (Biorad, Marnes la coquette, France) in a boiling water bath until complete dissolution of the protein precipitate. The reduced protein samples were then loaded (11μg per well) and separated by migration onto SDS-PAGE gels (*Mini-PROTEAN® TGX*™ Precast Gel 4–15% Biorad. This migration was carried out concomitantly with those of a molecular weight standard (BIOLINE, HyperPAGE, France) and a PTGS2 electrophoresis standard as a control for the PTGS2 molecular weight (69 kDa). Once the migration was completed, the gels were transferred onto 0.22 μm PVDF membranes (Biorad) in a semi-dry system (Trans-Blot® Turbo Transfer System, Biorad), at 2.5 A, 25 V for 7 minutes. Membranes were then washed in a solution of 50 mM Tris base, 137 mM NaCl and 0.1% Tween-20 (TBST 0.1%). Membrane saturation was performed in 0.1% TSBT with 5% milk (TBST-milk) for 2 h at room temperature under gentle agitation. PTGS2 protein was detected with a mouse monoclonal antibody

(Mab94) directed against ovine PTGS2 [35]. Incubation with the primary antibody was performed at a final concentration of 2 µg/ml in TBST-milk overnight at 4˚C under gentle agitation. The membranes were then washed twice for 10 minutes in TBST-milk and then twice for 10 minutes in TBST 0.1%. The secondary antibody (peroxidase donkey anti-mouse igG; Uptima, Interchim, Montluçon, France) was diluted to 1:12000 with TBST 0.1% and incubated with the membranes under stirring in the dark for 1 hour at room temperature.

Immunoreaction signals were revealed with the Pierce® ECL2 Western blotting Substrate according to the manufacturer's instructions (Thermo Fisher Scientific, Illkirch, France). The membranes were read using dedicated equipment in chemiluminescence and fluorescence mode (Camera CCD FUJIFILM LAS1000+; Raytest, Courbevoie, France).

**Immunolocalization.** Cryomatrix embedded samples from the upper portion of uterine horns were sectioned in 7 µm slices using a CM1950 cryostat (Leica Microsystems SAS, Nanterre, France). Upper portions were selected because the tissue structures were better preserved. The slices were collected on microscope slides (Superfrost® Plus, Menzel-Gläser, Thermo Fisher Scientific, Illkirch, France) and were incubated overnight at 4˚C with a blocking buffer (PBS 1x, donkey serum 1%, saponin 0.05%, bovine serumalbumin 2%). Sections were then incubated in a humidified atmosphere at room temperature with the primary antibody directed against ovine PTGS2 (final concentration: 2.89 µg/ml, mouse monoclonal antibody; Mab94; [35]). After three washes of 15 minutes with 0.1M PBS containing 0,2% BSA, the sections were incubated for 1 hour at room temperature, in darkness, and in a humidified atmosphere, with a a donkey anti-mouse FITC secondary antibody previously diluted at 1:200 in 0.1M PBS containing 0,2% BSA (Interchim, Montluçon, France). Sections were washed twice for 15 minutes in 0.1M PBS with 0,2% BSA and once in 0.1 M PBS. Finally, sections were covered with Vectashield Aantifade Mounting Medium containing 4′,6-diamidino-2-phenylindole (DAPI, Vector laboratories, Clinisciences, Nanterre, France). As a negative control, sections were incubated with the secondary antibody in the absence of the primary antibody. As a control of staining specificity, we used a section of ovine endometrium sampled at day 14 of gestation. Images were captured using confocal microscopy (Zeiss LSM700; MIMA2 plateform, INRAE IdF Jouy en Josas Antony, UMR BREED) and the Zen 2011 software (Zeiss, Marly le roi, France).

**Statistical analyses.** Statistical analyses and graphs were performed with GraphPad Prisma 7 and Microsoft Excel. A non-parametric Wilcoxon test with pairing was used to compare CAR and ICAR regions. A 2-factor ANOVA test (with pairing) was used to compare CAR and ICAR within each group. To analyze the difference in gene and protein expression between the two horns of each ewe, a Wilcoxon's test on expression means in each horn with a pairing when possible (more than two samples per variable), otherwise a test of Mann-Whitney was used. We compared the different groups two by two using a 2-factor ANOVA test with pairing on the uterine horns. For other results, Mann-Whitney test was used.

Differences were considered significant at the 95% confidence level ($p \leq 0,05$). Quantitative data were expressed as a median [1st Quartile-3rd Quartile] ±SD (standard deviation).

# Results

## Surgical results

Three UTx were performed with 2 veins (group 1) and 4 UTx with 1 vein (group 2). A control group was made of 3 simple hysterectomies. There was no significant difference in mean uterine artery diameter (4.5 mm (IQ 4.0–5.0) vs 4.8 mm (IQ 4.0–5.0), p = 0.61) nor in mean utero-ovarian vein diameter between right and left horns (7.8 mm (IQ 6.5–9.0) vs 6.4 mm (IQ 5.0–7.5), p = 0.12). Mean total operative time (608 min ±18 min vs 624 min ±89 min, p = 0.98),

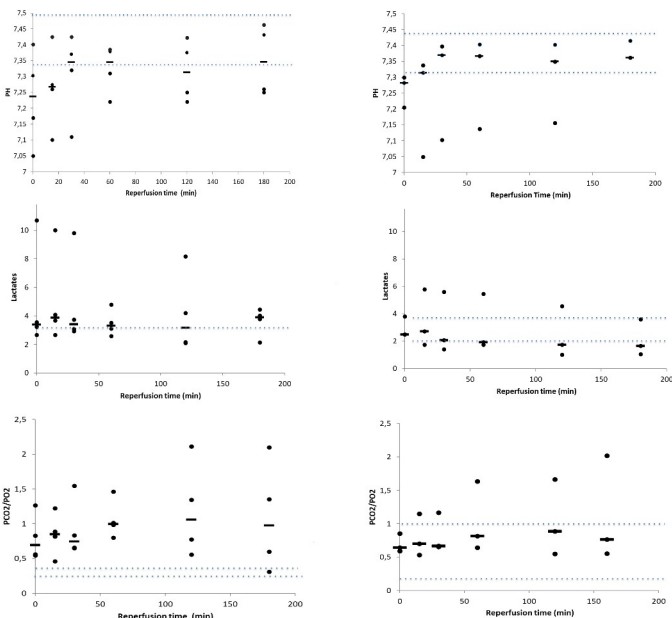

**Fig 4. Time course of pH, gas pressure and lactates in venous samples of transplanted uterus.** Blood was collected at various time points during the 180 min perfusion time. a) pH; b) pCO2/pO2; c) lactate. Dotted lines indicate the maximum and minimum range in samples taken before harvesting the graft. Individual values (rounds) and medians (line) are shown.

warm ischemia (74 min ±6 min vs 116 min ±16 min, p = 0.99) and cold ischemia (106 min ±63 min vs 131 min ±19 min, p = 0.24) were similar between groups 1 and 2.

## Blood gases

We found no statistically significant difference for pH, lactates or pCO2/pO2 between groups 1 and 2 during the time course of the UTx procedure (Fig 4). For group 1, blood gas median measures at 180 min were in between minimum and maximum ranges of samples taken before harvesting the uterus. For the group 2 they are hereafter. We note a trend toward an increase in pH and lactates during the first 30 min after reperfusion and then a stabilization beyond the first 30 minutes.

## Histology

No sign of apoptosis, stasis or tissue necrosis was observed in the various tissue layers of the uterus (endometrium, myometrium and perimetrium) in groups 1, 2 or controls. Endometrial infiltration by neutrophils was noticed (Fig 3) in all groups (Fig 5) without significant statistical difference between group 1 and 2 nor with control group. The various parameters estimated by stereology did not differ significantly between the three groups (Table 2).

**Quantification of gene expression in the endometrium.** In the endometrium, transcript levels were quantified for a selection of genes known to be involved in the regulation of (i) inflammation (*PTGS2*; superoxide dismutase, *SOD2*), (ii) apoptosis (*BAX* as a pro-apoptotic gene; *BCL2* as an anti-apoptotic gene), and (iii) immune function (complement component *C3*; interleukin-6; *IL6*; interleukin-8, *IL8*; Toll Like Receptor 4, *TLR4*) in transplanted solid organs.

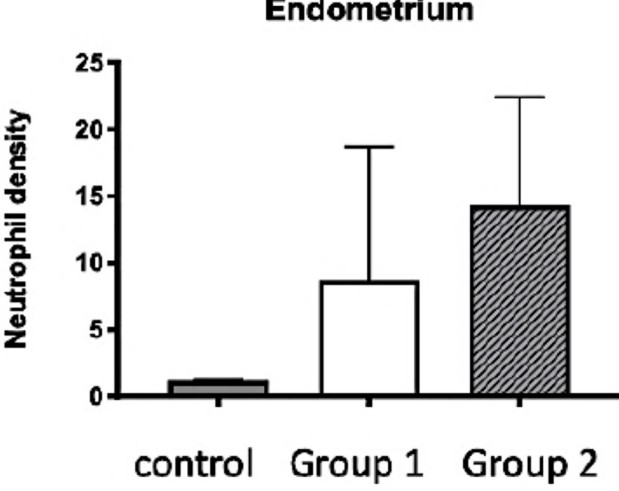

**Fig 5. Neutrophils density in the endometrium of auto-transplanted uterus.** Median density +/- SEM is presented for each phenotype (control group, group 1: Reperfusion of 2 veins, group 2: Reperfusion of 1 vein). For every ewe, median density of upper and lower part of the horns was calculated.

For each vascular phenotype, each horn of the uterus was defined according to its vascularization by indicating if the artery (A) and the efferent veins (V) were reperfused (noted as 1) or not reperfused (noted as 0). Regardless of the endometrial area (CAR or ICAR), there was no significant difference in gene expression levels between the 2 horns for any of the experimental groups (controls, group 1 and 2).

Nevertheless, transcript levels of the inflammation-related gene *PTGS2* and oxidative stress related gene *SOD2* were significantly higher in the ICAR areas compared with the CAR areas ((Fig 6a); *PGTS2*, fold-ratio = 1.84; $p < 0.0001$; *SOD2*, fold-ratio = 1.46; $p < 0.001$). Similarly, the *BAX/BCL2* ratio was 2.54 higher in the ICAR areas compared with the CAR areas ($p < 0.0001$). For the immune-related genes (*C3, IL6, IL8, TLR4*; Fig 6b) *C3* mRNA expression was 2.23 higher in the CAR areas compared with the ICAR regions ($p < 0.0001$) whereas *IL6* and *IL8* expression levels were significantly higher in the ICAR areas compared with the CAR areas (fold-ratio = 2.02, $p < 0.0001$ and fold-ratio = 2.59, $p = 0.001$ respectively). No significant difference was seen for *TLR4* transcript levels between the CAR and ICAR areas.

For each of the studied genes (Fig 6), expression levels in CAR or ICAR areas did not significantly differ between group 1 and group 2. In the [group 1 vs. control group] comparison,

**Table 2. Stereology: Histological structures (volume %) comparison of medians [1st Quartile-3rd Quartile] of upper horns between Group 1 (2 veins) Group 2 (1 vein) and control group.**

|  | Group 1 | Group 2 | Control |
|---|---|---|---|
| Epithelium | 2,62 [1.47–3.77] | 1.05 [0.69–3.46] | 0,75 [0.3–3.57] |
| Stroma | 31.15 [28.41–33.88] | 26.18 [20.59–43] | 32.35 [24.03–36.81] |
| Muscle | 23.57 [23.08–24.06] | 27.97 [23.31–42.15] | 29.22 [22.7–35.82] |
| Perimetrium | 10,56 [10.18–10.94] | 6.54 [4.85–9.95] | 4,91 [2.98–9.02] |
| Edema | 10.25[7.62–12.88] | 8.54[5.76–11.03] | 13.98 [8.23–19.71] |
| Artery | 0; 64 [0.36–0.92] | 0.86[0.51–1.37] | 1.53 [0.77–1.88] |
| Vein | 1.5 [0.33–2.77] | 2.2 [0.91–3.13] | 1.6 [0.98–3.24] |
| Muscular connective tissue | 12.44 [6.63–18; 24] | 7.38 [2.97–14.23] | 6.41 [3.58–10.02] |
| Gland | 7.2 [6.7–7.86] | 5.66 [4.07–7.15] | 5.06 [3.14–8.68] |

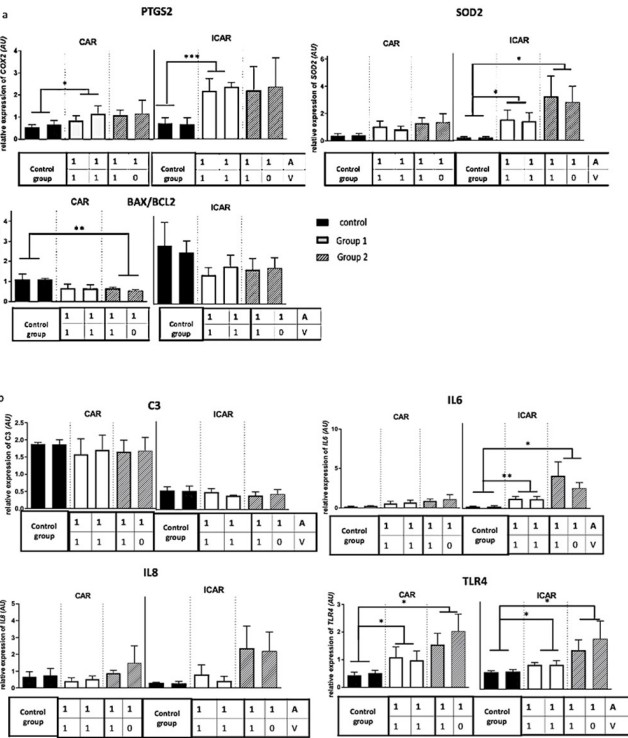

**Fig 6. Expression levels of a selection of genes in the endometrium of ovine auto-transplanted uteri.** (*a*) Selection of genes involved in inflammation, apoptosis, (*b*) selection of immune-related genes. For each group, mean expression and standard error of the mean (SEM) were presented as histograms. No surgery of the uterine vascularization was carried out in the control group (3 ewes). In groups 1 (3 ewes) and 2 (4 ewes), uterine horns were identified according to their vascularization; A = artery; V vein; 1 = reperfusion; 0 = absence of reperfusion. *, p<0.05; **, p<0.01, ***, p<0.001. AU: Arbitrary unit.

expression of *PTGS2* and *TLR4* transcripts was significantly higher in the CAR areas of group 1, while *PTGS2*, *SOD2*, *TLR4* et *IL6* transcript levels were significantly higher in the ICAR areas of group 1.

When the experimental group 2 was compared with the control group, in the CAR areas, *TLR4* expression level was significantly higher in the group 2, while the *BAX/BCL2* ratio was significantly lower. In the ICAR areas, *SOD2*, *TLR4* and *IL6* transcript levels were significantly higher in group 2.

## Expression and localisation of the PTGS2 protein in the endometrium of auto-transplanted uteri

Using ICAR endometrial samples (Fig 7 and S1 Fig), levels of ovine PTGS2 protein were estimated by Western Blot. There was a trend towards an increase in PTGS2 protein level in group 1 compared with group 2 (p = 0.28). There was no significant difference in PTGS2 protein expression between the two horns of each uterus.

Using a specific anti-ovine PTGS2 antibody (Mab94), cell localization of PTGS2 was investigated in the endometrium. Each vascular phenotype is presented with sections cut from endometrium collected from a representative ewe.

In keeping with former data that reported PTGS2 localization in ovine pregnant endometrium [35], PTGS2 was detected (i) in the cytoplasm and especially in the perinuclear part of cells, and (ii) in the luminal epithelial cells and subluminal glandular cells of the endometrium

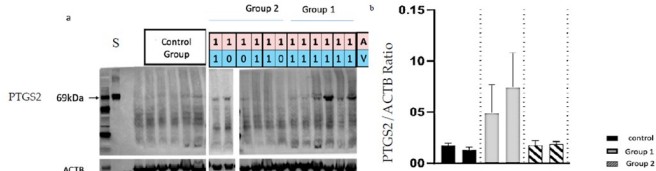

**Fig 7. Quantification of PTGS2 protein by Western Blot in the endometrium of auto-transplanted ovine uteri.**
The amount of PTGS2/protein was normalized to that of actin B protein (ACTB). For each group (control, group 1 and group 2), three ewes were analyzed. A = artery; V vein; 1 = reperfusion; 0 = absence of reperfusion. S: PTGS2 Standart.

sampled from a pregnant ewe at day 14 of gestation. A similar profile of PTGS2 cell distribution was seen in the endometrium of the uteri collected from the control group and auto-transplanted groups. No difference in the type of PTGS2-expressing cells was observed between the two horns, regardless of the group (control, group 1, group 2). Nevertheless, PTGS2 labelling intensity appears to be the most intense in the endometrium of control group compared with group 1 and 2, and the lowest in the endometrial tissue sampled from group 2 uteri (Fig 8).

## Discussion

Being the first and unique example of time-limited organ transplantation, UTx aims at improving the quality of life by offering the possibility of live birth to otherwise permanently infertile women. In an effort to reduce the risks for both LDs and recipients and shorten the duration of surgery while improving overall UTx success rates, our team has worked on possible surgical simplifications such as using utero-ovarian rather than uterine veins as venous outflow with reported births of healthy babies [9,11]. Using only one vein could avoid bilateral dissection for the LD and thereby decrease complications and shorten warm ischemia time for the recipient. In case of unilateral venous thrombosis, data about graft recovery and function are missing. According to Moore's criteria and the IDEAL concept (**I**dea, **D**evelopment, **E**xploration, **A**ssessment, **L**ong term study), we used preclinical research in the animal model before introducing a novel surgical strategy [29,30].

At the time of transplantation, the organ is exposed to various risks, which may have long-term consequences. The duration of ischemia has been singled out as having an impact on acute and/or chronic rejection in kidney transplantations, eventually leading graft failure [25,26]. The role of ischemia and its duration have been incriminated in reperfusion cellular damages such as inflammation, cellular death and oxidative stress, which have been reported in kidney, liver, lung and cardiac transplantations [26,27,36,37]. Previous animal trials in UTx investigated early ischemia-reperfusion markers including blood gases and histology as well as long term consequences on pregnancy issues using imaging and histology [31,38]. In the present study, levels of blood gases (pH, pO2/pCO2 and lactates) or histological analyses (signs of necrosis, apoptosis, edema and endometrial neutrophil density) did not differ between the group 1 (anastomosis of two utero-ovarian veins) and 2 (anastomosis of one utero-ovarian vein) after a 3-hour reperfusion time. Although the values were heterogeneous between animals, the levels of lactates and pH gradually improved to stabilize starting 30 min post-grafting with values of pCO2/pO2 stabilizing starting after 60 min, which is consistent with the literature [31]. This demonstrates satisfactory organ recovery over a short-term period. The time of reperfusion and ischemia being relatively short in our series, this probably explains why no signs of necrosis, stasis, apoptosis or oedema was detected by histological examination nor by stereology. These signs are observed with a longer period of ischemia which is more likely

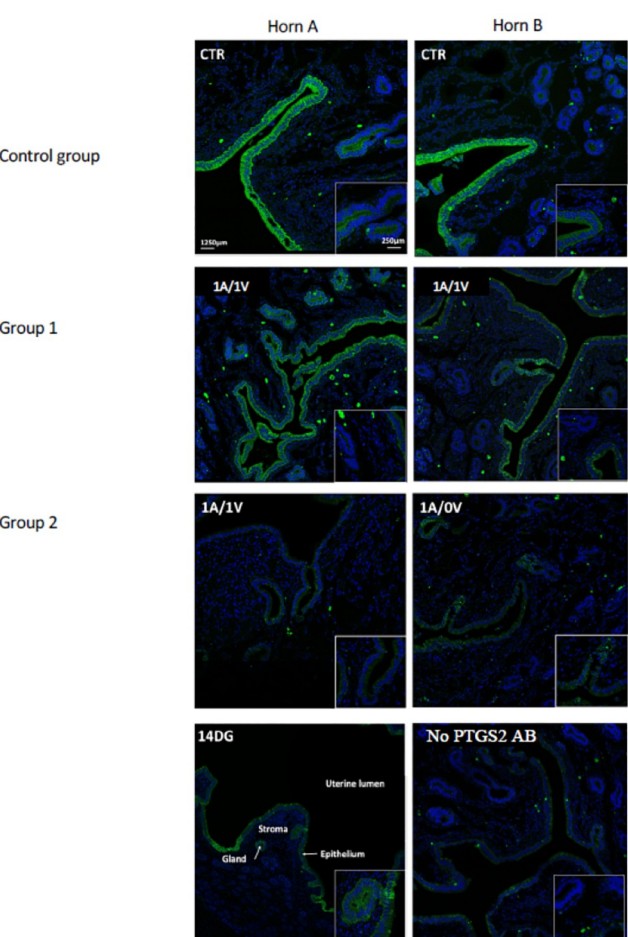

**Fig 8. Immunolocalization of PTGS2 protein in the endometrium of auto-transplanted ovine uteri.**
Immunohistochemistry was performed using sections cut from ovine endometrium. For each uterus, the two horns
are labelled horn A and horn B to make the interpretation easier. Uterine vascularization is indicated in the top left
corner of each microphotograph. CTR: Control; A = artery; V vein; 1 = reperfusion; 0 = absence of reperfusion. No
surgery of the uterine vascularization was carried out in the control group. Specificity of ovine PTGS2 labelling was
carried out using (i) a section of ovine endometrium sampled from a ewe at day 14 of gestation (14DG), and (ii) a
section of endometrium sampled from a control uterus incubated in the absence of the primary antibody (No PTGS2
AB). Bars = 1250 μm (insert: 250 μm).

occur in transplantation models with deceased donor and after long-time of cold ischemia
[39]. Neutrophils infiltration of the endometrium—an early marker of inflammation—was
observed in all cases, confirming previous data in sheep after auto transplantation [40]. Yet, no
significant difference in neutrophils number was observed between the groups 1 and 2 veins.

Few studies have addressed the UTx process at the molecular level. Expression of genes
involved in inflammation and immunity including *PTGS2*, *IL6*, *IL8*, *TLR4*, *C3* and *SOD2* have
been established in others SOT [41–44]. In our present study, the expression levels of *PTGS2*,
*IL6*, *SOD2* and *TLR4* transcripts were higher in the ICAR and/or CAR regions of the reper-
fused endometrium in one and two vein groups compared with the control endometrium.
Therefore, auto-transplantation and then reperfusion are associated with the altered expres-
sion of transcripts of genes involved in inflammation and immunity. Cyclooxygenase-2
(*PTGS2* transcript) is an enzyme induced by inflammation involved in prostaglandin H2 pro-
duction [45]. Interleukine-6 (IL6) is a pro-inflammatory cytokine induced by inflammation

which stimulate antibodies and B lymphocytes production [41].*PTGS2* as well as *IL6* have been shown to be involved during the acute phase of the ischemia-reperfusion process in kidney or liver transplantation in different animal models and in humans [45,46]. Superoxide Dismutase (SOD2) is a mitochondrial enzyme involved in neutralizing reactive oxygen Species (ROS), including the superoxide ion [47,48]. *SOD2* gene expression is increased in solid organ ischemia [44]. Toll Like Receptor 4 (TLR4) is a transmembrane receptor with a role in innate immunity. It is expressed in immune cells (monocytes, neutrophils, lymphocytes) as well as parenchymal cells (kidney, liver, lung, skin) [49]. Higher expression of *TLR4* is associated with increased time to recovery of function of the grafted organ in renal transplantation as well as acute and chronic rejection in renal and liver transplantation [43]. BAX is a pro-apoptotic gene whose protein induces the release of mitochondrial proteins located between the outer and inner membranes of the mitochondria, in particular cytochrome c, Bcl2 is an anti-apoptotic gene whose protein inhibits the activation of BAX. The ratio of pro-apoptotic to anti-apoptotic proteins is the determinant of cell survival or death [50]. BAX/BCL2 ratio has been studied in liver and kidney transplantation [44,51] and recently in UTx [52]. Based on TUNEL assay caspase-3 immunolocalization [39,53] and PCR quantification of *BCL2* and *BAX* transcript levels, no apoptosis was detected in the ovine uterus 1 day after auto-transplantation in the ewe [52]. Our results are consistent with these data since no sign of apoptosis was detected in the uterine tissue using histological assessment or gene quantification of *BAX* and *BLC2*. Protein C3 complement(C3) is involved in the alternative pathway of activation of the complement cascade. Once cleaved, fragment C3a will attract inflammatory cells (neutrophils, macrophages and monocytes) to the site of complement activation while fragment C3b facilitates phagocytosis. No increase in *C3* expression levels was detected in UTx. The impact of complement in ischemia-reperfusion lesions has been demonstrated in the kidney, heart and intestine [54]. Interleukine-8 (IL8) is a chemokine that attracts polynuclear cells at the site of its secretion. Contrary to previous reports [31], we did not observe a significant increase in neutrophils density and in IL8 mRNA expression in auto-transplantation, either with one, or two veins outflow compared with control group. Yet, a trend was observed, which did not reach statistical significance due to the small number of cases studied. In lung transplantation, IL8 protein expression 2 hours after reperfusion was correlated with the duration of ischemia, with lung function assessed by the Pa(O2)/FI(O(2)) ratio, the mean airway pressure, and the APACHE score during the first 24 postoperative hours and the length of hospitalization in intensive care [42]. UTx was not associated with an histological alterations. Histological analyses of the uterine tissues have already unveiled the absence of impact following auto-transplantation in the ovine model [31]. Evaluation of a selection of transcripts such as *PTGS2*, *BAX/BCL2*, *C3*, *IL6*, *IL8*, *SOD2* and *TLR4* could be more relevant as ischemia-reperfusion early markers as shown in our study. Following-up these markers over time will be required to establish whether an early increase of these markers related to inflammation and immunity is correlated with the recovery of the graft functionality, as it has been described in other SOT [43].

Expression of *PTGS2*, *SOD2*, *IL6* and *IL8* mRNA was significantly higher in the ICAR areas of the endometrium in auto-transplanted groups (1 and 2). In sheep, the ICAR endometrial areas contains the uterine glands whereas the CAR endometrial areas are aglandular. Our findings are consistent with the cellular localization of PTGS2 observed in the luminal and glandular epithelium of the ovine endometrium [35]. The ratio of *BAX*/*BCL2* ratio was also higher in the ICAR areas compared with the CAR regions. These data indicate a rise in inflammation and innate immunity in the ICAR areas as well as a relative increase in apoptosis-promoting genes compared to anti-apoptotic genes. The ICAR areas, the localization of endometrial glands, therefore appear to be more sensitive to ischemia-reperfusion. To our knowledge, the difference between CAR and ICAR areas has not been taken into account in previous UTx

studies in Sheep [39,52]. By investigating the endometrial glandular areas, our study underlines that ICAR areas appear more sensitive to ischemia-reperfusion. Receptive and synchronized endometrium is necessary for implantation and pregnancy [55]. In ewes, PTGS2 protein expression is variable across the oestrus cycle: Undetectable on the day post-oestrus(D3), the PTGS2 protein is transiently expressed during the luteal phase, between D12 and D15. Immunolocalization of PTGS2 protein in the luminal and glandular epithelium of glands located in the vicinity of the uterine lumen is consistent with literature [35]. Cell labelling was comparable between the two horns in keeping with the absence of significant differences between the horns for *PTGS2* mRNA and protein levels. In the control group, the uteri underwent as little ischemia as possible. Detection of PTGS2 signal in endometrial cells in this group may be a consequence of progesterone released by the vaginal pessary used to homogenize progesterone levels between ewes before surgery [35].

## Conclusion

Based on the molecular, cellular and histological parameters that were investigated, our data have shown that anastomosis of a single uterine vein or two uterine veins have similar effects on uterine tissues upon a 3 hours reperfusion time. There was no significant difference in transcript levels of ischemia-reperfusion markers (*PTGS2*, *BAX/BCL2*, *C3*, *IL6*, *IL8*, *SOD2* and *TLR4*) nor in PTGS2 protein expression and cell localization. More importantly, no significant difference was found between the two uterine horns collected from the ewes of the experimental group with one venous outflow. This suggests the presence of vascular anastomosis between the two horns that takes place at the early stages of the reperfusion process. Considering the early steps of reperfusion, our data suggest that one vein could be sufficient for uterus transplantation. Based on our ovine experimental model, additional analyses will be needed to study the long-term time-course of reperfusion in order to confirm that anastomosis of one single vein (either at transplantation or as a consequence of a unilateral venous thrombosis occurring during the post-reperfusion process) allows that the transplanted uterus supports pregnancies with the birth of viable and healthy lambs.

## Supporting information

**S1 Fig. The original uncropped all blot results for PTGS2 and ACTB.** R: right horn, L: left horn.
(TIF)

## Acknowledgments

Foundation Foch.
MIMA2 platform.
@BRIDGeplatform (Agilent Analyzer, INRAE, Jouy-en-Josas, France).
Pr De Ziegler and Claire Marchiori for Proof reading.

## Author Contributions

**Conceptualization:** Marie Carbonnel, François Vialard, Pascale Chavatte-Palmer, Olivier Sandra, Jean-Marc Ayoubi.

**Data curation:** Marie Carbonnel, Nathalie Cornet.

**Formal analysis:** Marie Carbonnel, Nathalie Cornet.

**Funding acquisition:** Jean-Marc Ayoubi.

**Investigation:** Marie Carbonnel, Nathalie Cornet, Angéline Favre-Inhofer, Laurent Galio, Mariam Raliou, Anne Couturier-Tarrade, Corinne Giraud-Delville, Gilles Charpigny, Val-érie Gelin, Olivier Dubois, Barbara Hersant, Romain Bosc, Raphael Coscas, Christophe Richard.

**Methodology:** Marie Carbonnel, Aurélie Revaux, Pascale Chavatte-Palmer, Olivier Sandra, Jean-Marc Ayoubi.

**Project administration:** Christophe Richard.

**Resources:** Gilles Charpigny.

**Supervision:** François Vialard, Pascale Chavatte-Palmer, Olivier Sandra, Jean-Marc Ayoubi.

**Validation:** Marie Carbonnel, Nathalie Cornet, Aurélie Revaux, Angéline Favre-Inhofer, Lau-rent Galio, Mariam Raliou, Anne Couturier-Tarrade, Corinne Giraud-Delville, Gilles Char-pigny, Valérie Gelin, Olivier Dubois, Barbara Hersant, Romain Bosc, Raphael Coscas, François Vialard, Christophe Richard, Olivier Sandra, Jean-Marc Ayoubi.

**Visualization:** Marie Carbonnel, Nathalie Cornet, Aurélie Revaux, Angéline Favre-Inhofer, Laurent Galio, Mariam Raliou, Anne Couturier-Tarrade, Corinne Giraud-Delville, Gilles Charpigny, Valérie Gelin, Olivier Dubois, Barbara Hersant, Romain Bosc, Raphael Coscas, Pascale Chavatte-Palmer, Christophe Richard, Olivier Sandra, Jean-Marc Ayoubi.

**Writing – original draft:** Marie Carbonnel.

**Writing – review & editing:** Marie Carbonnel, Nathalie Cornet, François Vialard, Pascale Chavatte-Palmer, Olivier Sandra, Jean-Marc Ayoubi.

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
