## [Decision Letter · Decision Letter 0]

26 Feb 2021

PONE-D-21-00898

Analysis of blood parameters and molecular endometrial markers during early reperfusion in two ovine models of uterus transplantation

PLOS ONE

Dear Dr. Carbonnel,

Thank you for submitting your manuscript to PLOS ONE. After careful consideration, we feel that it has merit but does not fully meet PLOS ONE’s publication criteria as it currently stands. Therefore, we invite you to submit a revised version of the manuscript that addresses the points raised during the review process.

We look forward to receiving your revised manuscript.

Kind regards,

Antonio Simone Laganà, M.D., Ph.D.

Academic Editor

PLOS ONE

Journal Requirements:

3.We note that you have indicated that data from this study are available upon request. PLOS only allows data to be available upon request if there are legal or ethical restrictions on sharing data publicly. For information on unacceptable data access restrictions, please see http://journals.plos.org/plosone/s/data-availability#loc-unacceptable-data-access-restrictions.

5.Thank you for stating the following financial disclosure:

 "no"

6. Thank you for stating the following in your Competing Interests section: 

"no"

7. Please upload a new copy of Figure 3, 4, 6a and 6b as the detail is not clear. Please follow the link for more information: https://blogs.plos.org/plos/2019/06/looking-good-tips-for-creating-your-plos-figures-graphics/" https://blogs.plos.org/plos/2019/06/looking-good-tips-for-creating-your-plos-figures-graphics/

Additional Editor Comments:

The topic of the manuscript is interesting. Nevertheless, the reviewer raised several concerns: considering this point, I invite authors to perform the required major revisions.

Reviewers' comments:

Reviewer's Responses to Questions

**Comments to the Author**

1. Is the manuscript technically sound, and do the data support the conclusions?

Reviewer #1: Partly

2. Has the statistical analysis been performed appropriately and rigorously? 

Reviewer #1: I Don't Know

3. Have the authors made all data underlying the findings in their manuscript fully available?

Reviewer #1: No

4. Is the manuscript presented in an intelligible fashion and written in standard English?

Reviewer #1: No

5. Review Comments to the Author

Reviewer #1: Below is the review for MS titled “Analysis of blood parameters and molecular endometrial markers during early reperfusion in two ovine models of uterus transplantation” which was submitted to PLOS One. The researchers wanted to provide insight on the influence of one or two vein anastomosis on the early steps of ischemia-reperfusion in an ovine model of uterine transplantation.

While the study is interesting, some concerns remain and are listed below. Overall, the MS would be greatly strengthened by a thorough editing for English grammar, sentence structure, and tenses of verbs etc. Some sentences are difficult to understand as written. For example, to name a few, lines 375 -376; Lines 473-474 (what is meant by a high quality endometrium?) Lines 480-481.

In the introduction a brief narrative on the rationale and justification for uterine transplants would reinforce the significance of the MS. Currently, it is not clear why transplants are needed or performed. How common is this?

For the western blots assay, how much protein was loaded per well? Please include.

How was neutrophil density quantitated? Please describe.

The figure legends appear out of place in the MS and it was very difficult to ascribe specific legends to the appropriate figure when reviewing. Further all the figures are very blurry, even when the JPEG pictures were downloaded. It was difficult to read figure axis and labels etc. for all. Figure 4 needs thorough editing as data are not legible or understandable as presented.

6. PLOS authors have the option to publish the peer review history of their article (what does this mean?). If published, this will include your full peer review and any attached files.

Reviewer #1: No

---

## [Author Response · Author response to Decision Letter 0]

2 Apr 2021

REBUTTAL LETTER

Responses to Reviewer #1

Overall, the MS would be greatly strengthened by a thorough editing for English grammar, sentence structure, and tenses of verbs etc. Some sentences are difficult to understand as written. For example, to name a few, lines 375 -376; Lines 473-474 (what is meant by a high-quality endometrium?) Lines 480-481.

The manuscript was edited by a fluent English speaker.

- We clarified some sentences. For example: LINE 356-358. “Nevertheless, PTGS2 labelling intensity appears to be the most intense in the endometrium of control group compared with group 1 and 2, and the lowest in the endometrial tissue sampled from group 2 uteri (Fig 8).”459-460. “Receptive and synchronized endometrium is necessary for implantation and pregnancy [55]”Lines 466-468 “Detection of PTGS2 signal in endometrial cells in this group may be a consequence of progesterone released by the vaginal pessary used to homogenize progesterone levels between ewes before surgery[35].” 

In the introduction a brief narrative on the rationale and justification for uterine transplants would reinforce the significance of the MS. Currently, it is not clear why transplants are needed or performed. How common is this?

- We added in the introduction the rationale and justification for uterine transplantation: “In vitro fertilization (IVF), developed over 40 years ago for treating tubal infertility, today is used to bypass nearly all infertility issues. However, one cause of infertility remains untreated: uterine factor infertility (UFI). UFI, which is estimated to affect 1-5% of women, can be congenital (complete uterine agenesis or other severe congenital malformations) or acquired (from hysterectomy or loss of a functional endometrium)[1]”. Line 41-44.

For the western blots assay, how much protein was loaded per well? Please include.

Eleven (11) µg protein were loaded per well. This precision has been added in the “Material and Methods” section (LINE 210).

How was neutrophil density quantitated? Please describe.

The density of neutrophils was quantified by counting tissue bound neutrophils in 10 area that were randomly chosen (x40 magnification), with the use of a grid. This description of the method has been added in the “Material and Method” section (LINE 158-160)

The figure legends appear out of place in the MS and it was very difficult to ascribe specific legends to the appropriate figure when reviewing. Further all the figures are very blurry, even when the JPEG pictures were downloaded. It was difficult to read figure axis and labels etc. for all. Figure 4 needs thorough editing as data are not legible or understandable as presented.

We apologize for any inconvenience in reading our data. The quality of figures and graphs has been improved in the revised version of our manuscript. Legends have been ascribed to the figures.

---

## [Decision Letter · Decision Letter 1]

28 Apr 2021

Analysis of blood parameters and molecular endometrial markers during early reperfusion in two ovine models of uterus transplantation

PONE-D-21-00898R1

Dear Dr. Carbonnel,

We’re pleased to inform you that your manuscript has been judged scientifically suitable for publication and will be formally accepted for publication once it meets all outstanding technical requirements.

Kind regards,

Antonio Simone Laganà, M.D., Ph.D.

Academic Editor

PLOS ONE

Additional Editor Comments (optional):

Authors performed the required corrections, which were positively evaluated by the reviewers. I am pleased to accept this paper for publication.

Reviewers' comments:

Reviewer's Responses to Questions

**Comments to the Author**

1. If the authors have adequately addressed your comments raised in a previous round of review and you feel that this manuscript is now acceptable for publication, you may indicate that here to bypass the “Comments to the Author” section, enter your conflict of interest statement in the “Confidential to Editor” section, and submit your "Accept" recommendation.

Reviewer #1: All comments have been addressed

2. Is the manuscript technically sound, and do the data support the conclusions?

Reviewer #1: Yes

3. Has the statistical analysis been performed appropriately and rigorously? 

Reviewer #1: Yes

4. Have the authors made all data underlying the findings in their manuscript fully available?

Reviewer #1: Yes

5. Is the manuscript presented in an intelligible fashion and written in standard English?

Reviewer #1: Yes

6. Review Comments to the Author

Reviewer #1: (No Response)

7. PLOS authors have the option to publish the peer review history of their article (what does this mean?). If published, this will include your full peer review and any attached files.

Reviewer #1: No

---

## [Editor Report · Acceptance letter]

3 May 2021

PONE-D-21-00898R1 

Analysis of blood parameters and molecular endometrial markers during early reperfusion in two ovine models of uterus transplantation 

Dear Dr. Carbonnel:

I'm pleased to inform you that your manuscript has been deemed suitable for publication in PLOS ONE. Congratulations! Your manuscript is now with our production department. 

Kind regards, 

on behalf of

Dr. Antonio Simone Laganà 

Academic Editor

PLOS ONE